

# Freeze-like responses to pain in humans and its modulation by social context

Kai Karos[1,2], Ann Meulders[2,3], Tine Leyssen[3] and Johan W. Vlaeyen[2,3]

[1] Centre for the Psychology of Learning and Experimental Psychopathology, Faculty of Psychology and Educational Sciences, KU Leuven, Leuven, Vlanders, Belgium
[2] Experimental Health Psychology, Clinical Psychological Science, Faculty of Psychology and Neuroscience, Maastricht University, Maastricht, Limburg, Netherlands
[3] Research Group on Health Psychology, Faculty of Psychology and Educational Sciences, KU Leuven, Leuven, Vlanders, Belgium

## ABSTRACT

**Background**. Maladaptive defensive responses such as excessive avoidance behavior have received increasing attention as a main mechanism for the development and maintenance of chronic pain complaints. However, another defensive response which is commonly studied in animals as a proxy for fear is freezing behavior. No research to date has investigated human freezing behavior in the context of pain. In addition, there is an increasing realization that social context can affect pain-relevant processes such as pain experience and pain behavior but less is known about the effects of social context on defensive responses to pain. Hence, this study investigated freezing behavior and facial pain expression in the context of pain, and their modulation by social context.

**Methods**. Healthy, pain-free participants ($N = 39$) stood on a stabilometric force platform in a threatening or safe social context, which was manipulated using angry or happy facial stimuli. In some trials, an auditory cue (conditioned stimulus; CS) predicted the occurrence of painful electrocutaneous stimulus (unconditioned stimulus; pain-US). We assessed body sway (an index of freezing), heart rate, facial pain expression, self-reported pain intensity, unpleasantness, and pain-US expectancy during the CS and the context alone (no CS).

**Results**. The results were mixed. Neither the anticipation of pain, nor social context affected body sway. Heart rate and painful facial expression were reduced in the threatening social context at high anxiety levels. A threatening social context also elicited higher pain-US expectancy ratings. In sum, a threatening social context increases the expectation of pain, but reduces the facial expression of pain and lowers heart rate in highly anxious individuals.

Corresponding author
Kai Karos, kai.karos@kuleuven.be

## INTRODUCTION

Pain is conceptualized and defined as a subjective response which is modulated by biological, psychological, and social factors (*De C Williams et al., 2016*; *Karos et al., 2018a*). While acute pain is adaptive in the short-term and promotes healing, chronic pain persists beyond time of healing and is associated with severe disability and reduction in quality
of life. Currently, chronic pain is a growing global health concern with one in five people in Europe and the US suffering from chronic pain (*Breivik et al., 2006*; *Kuehn, 2018*). Consequently, it is imperative to understand how chronic pain develops and is maintained.

One of the most influential models of chronic pain is fear-avoidance model (*Crombez et al., 2012*; *Vlaeyen & Linton, 2000*; *Vlaeyen & Linton, 2012*), which proposes that pain-related fear and persistent maladaptive defensive responses such as avoidance lead to chronic pain-related disability. For example, patients might be afraid and consequently avoid activities that they believe to be associated with pain (*Meulders et al., 2016*). But how do formerly adaptive responses become maladaptive? Growing evidence suggests that the social context can play an important role. A threatening social context can worsen the experience of painful stimuli (*Karos et al., 2019*; *Karos et al., 2018a*; *Krahé et al., 2013*), the communication of pain to others (*Karos et al., 2019*; *Karos et al., 2018*; *Peeters & Vlaeyen, 2011*), and facilitate the acquisition of pain-related fear (*Karos, Meulders & Vlaeyen, 2015*). More generally, chronic pain has been associated with a whole host of threatening social experiences such as social isolation and injustice (*Karos et al., 2018a*). In addition, early traumatic experiences of social threat such as bullying have been associated with increased risk for the development of chronic pain later on *Fekkes et al. (2006)* and *Voerman et al. (2015)*. Similarly, people who are marginalized by social conditions (e.g., refugees, less well educated, living in poverty) are especially at risk (*Craig et al., 2020*).

Notwithstanding, the proposed importance of maladaptive defensive responses in the etiology and maintenance of chronic pain complaints in contemporary fear avoidance models, there is little research evaluating the modulation of defensive responses to pain by social context (*Karos et al., 2018a*). Most research to date has focused on behavioral avoidance responses, which are hypothesized to be at the center of the development and maintenance of chronic pain complaints (*Meulders et al., 2016*; *Volders et al., 2015*). However, avoidance is not the only defensive response of interest.

Another defensive response to threat, which is commonly used as the main outcome measure for fear in animal studies, is freezing. It is primarily characterized by reduced body motion and bradycardia (decreased heart rate) (*Glombiewski et al., 2015*; *Hagenaars, Oitzl & Roelofs, 2014*), but has also been associated with changes in the sympathetic nervous system (e.g., increased arterial pressure, increased respiration, increased muscle tone) (*Roelofs, 2017*). While freezing has been studied in animal research for decades, research on human freeze-like behavior is only recently emerging (*Allcoat et al., 2015*; *Azevedo et al., 2005*; *Facchinetti et al., 2006*; *Hermans et al., 2013*; *Mobbs et al., 2009*). Evidently, humans also show a freeze-like response (reduced body sway and bradycardia) when exposed to threatening films (*Hagenaars, Roelofs & Stins, 2014*). Especially relevant for the current study, another study demonstrated increased freeze-like behavior in humans in response to social threat (i.e., angry facial stimuli). Interestingly, this behavior was especially pronounced in highly anxious individuals (*Roelofs, Hagenaars & Stins, 2010*). This study was recently partly replicated, demonstrating that freezing occurs not only in response to physical threat but also social threat and is modulated by anxiety (*Noordewier, Scheepers & Hilbert, 2019*).

While adaptive in the short-term, freezing could become maladaptive when sustained and obstructing flexible adaptations to changes in the environment (*Buss et al., 2004*; *Hagenaars, Oitzl & Roelofs, 2014*; *Hagenaars et al., 2008*). There is some evidence that increased freezing responses are associated with psychopathology, such as in the case of aversive early life events (*Hagenaars, Stins & Roelofs, 2012*), increased state anxiety (*Roelofs, Hagenaars & Stins, 2010*), and increased nonspecific reduced mobility in patients with panic disorders (*Lopes et al., 2009*) but no such research exists within the domain of pain. Similarly, to maladaptive avoidance behavior, prolonged freezing behavior could increase the risk for the development of chronic pain (e.g., by facilitating physical immobility to perceived threat of pain) and a threatening social context could be one of the factors facilitating such a development.

Threat appraisal and associated defensive responses can directly affect pain reports (*Jackson et al., 2009*; *Jackson et al., 2005*; *Vlaeyen et al., 2009*). However, research into human freezing responses in the context of pain is scarce. Animal research indicates that rats freeze during the anticipation of painful electrical stimuli (*De Castro Gomes & Landeira-Fernandez, 2008*; *Rosen, 2004*). Moreover, findings in human research are indicative of freezing responses to pain. A study in patients with chronic low back pain showed that patients exhibited freeze-like behaviors when instructed to move their trunk as fast as possible (*Bourigua et al., 2014*). Another line of research demonstrated that pain behavior is reduced in a threatening social context. *Peeters & Vlaeyen (2011)* found that participants show less painful facial expression when receiving electrical stimuli in a threatening social context, possibly a result of freeze-like reductions of overall body movement. Subsequent attempts to replicate this finding have been mixed (*Karos et al., 2019*; *Karos et al., 2018*). However, no study to date has investigated the effect of pain on actual body sway and bradycardia, or its modulation by social context.

Consequently, the goal of this study was to investigate the effect of painful stimuli on markers of freezing, specifically body sway, heart rate, and facial expressions of pain. Moreover, we investigated whether freezing responses were modulated by social context and anxiety levels. To this end, we conceptually replicated the study by *Roelofs, Hagenaars & Stins (2010)*, applying it to the domain of pain. We hypothesized that the anticipation of pain would be associated with reduced body sway (Hypothesis 1), and that this effect would be more pronounced in a threatening social context compared to a safe social context and in highly anxious individuals (*Roelofs, Hagenaars & Stins, 2010*). We also hypothesized that a threatening social context overall is associated with bradycardia, especially so in highly anxious individuals (Hypothesis 2). In line with earlier research, we were also interested whether painful facial expressions in the threatening social context are reduced compared to a safe social context (Hypothesis 3), which could be a sign of overall bodily freezing.

## MATERIALS AND METHODS

### Participants

Thirty-nine healthy, pain-free individuals (31 females; mean age ± SD = 22.79 ± 3.07; range = 18–33) voluntarily participated in this study. Participants were recruited by means

of flyers and the departmental experiment management system (EMS; Sona Systems Ltd.). The majority of the participants were students ($n = 34$; 87%) and ten (25.6%) participants were working. Of the 39 participants, 34 were living alone (87.2%) and five (12.8%) coinhabiting with someone else. Regarding highest education, 16 (41%) participants had completed middle school, 10 (25.6%) participants high school, and 13 (33.3%) participants university education. The exclusion criteria were pregnancy, current or history of cardiovascular disease, chronic or acute respiratory disease (e.g., asthma, bronchitis), neurological diseases (e.g., epilepsy), any current or past psychiatric disorders, acute or chronic pain, uncorrected hearing problems, cardiac pacemaker or the presence of any other electronic, medical devices, impaired, uncorrected vision, or use of anxiolytics or antidepressants. All participants received a financial compensation of €15 for their participation. Power calculations were run using GPower (*Faul et al., 2007*), assuming a medium effect size of Cohen's f = .25, a (conservative) between-measurement correlation of .05, an alpha level of .05, and aiming for a power of 90%, resulting in a required sample of 36 participants. A medium effect size was based on the range of effect sizes found in the original study by *Roelofs, Hagenaars & Stins (2010)*.

## Ethical approval

The experimental protocol was approved by the Social and Societal Ethics Committee of the KU Leuven (Belgium) (registration number = S-56678). All participants provided written informed consent prior to participation. It was emphasized that participation was completely voluntary and that participants were allowed to stop the experiment at any time without any negative consequences.

## Apparatus and experimental stimuli
### Stabilometric force platform
Postural stability and body sway of each participant were assessed using a NeuroCom Clinical Research System (NeuroCom International, Inc., Clackamas, OR, USA) (see http://www.interempresas.net/FotosArtProductos/P102631.jpg). The system comprises two independent (23 × 46 cm) 6 degrees of freedom AMTI force plates and a three-sided surround. Vertical forces exerted on the plates were recorded at a sampling rate of 100 Hz and were used to derive the center of pressure (COP) time series for each participant in the anterior-posterior (AP), and medio-lateral (ML) directions. Participants wore a safety harness that was only engaged in the case of loss of balance. Note that the platform was stable at all times when participants stood on it. This relatively stable position enables a larger range of movement in the AP direction than in the ML direction, and therefore makes AP movements more susceptible than ML movements to affective modulations (*Roelofs, Hagenaars & Stins, 2010*). The platform also includes a 15-inch, height-adjustable computer screen, which was positioned approximately 40 cm in front of the participant adjusted to eye-level.

### Heart rate monitor

Heart rate was measured as beats per minute (bpm) with a Polar heart rate monitor (Polar Electro Oy, Kempele, Finland), which consists of an electrode belt and transmitter W.I.N.D and a wireless heart rate monitor RS800CX.

### Software

The entire experiment was run on a Windows XP computer (Dell Optiplex 755) with 2 GB RAM and an Intel Core 2 Duo processor at 2.33 GHz and an ATI Radeon 2400 graphics card with 256 MB of video RAM. Stimulus presentation was controlled by Affect (version 4.0) (*Spruyt et al., 2010*).

### Stimulus material

A painful electrocutaneous stimulus of 2 ms duration (single squared waveform pulse) with a 2000 μs pulse duration and a maximum voltage of 400V served as the unconditioned stimulus (pain-US) in the present experiment. The electrical stimulation was delivered by a commercial stimulator (DS7A, Digitimer, Welwyn Garden City, England) through surface SensorMedics electrodes (1 cm diameter) filled with K-Y gel that were attached to the wrist of the right hand of the participant. To select the intensity level of the pain-US, participants were repeatedly exposed to electrocutaneous stimulation of increasing intensity. They were asked to rate each stimulus on a scale ranging from 0 (feeling nothing) to 10 (worst pain imaginable). The participant was instructed to select a stimulus intensity with a rating of about 8, which was "*moderately painful and demanding some effort to tolerate*" (mean self-reported stimulus intensity was 8.08, $SD = 0.27$, range = 8–9). After selecting the pain stimulus the participant was informed that (s)he would repeatedly receive stimuli of the maximum calibrated intensity during the remainder of the experiment. They were also given the possibility to increase or decrease the selected stimulus intensity at this point (mean physical stimulus intensity was 32.75 mA, $SD = 20.55$, range = 2–84 mA).

Based on previous a previous pilot test, a 44,100 Hz auditory cue of 1s duration presented binaurally via headphones at 90 dBA (HD 202, Sennheiser) served as the conditioned stimulus (CS). Social context was manipulated using facial stimuli. The facial stimuli consisted of emotional faces taken from 20 models (10 male and 10 female) in the Karolinska Directed Emotional Faces database (*Lundqvist, Flykt & Öhman, 1998*). Each model showed two different affective expressions (happy and angry), resulting in a total of 40 stimuli. The stimuli were the same as used in the study by Roelofs, Hagenaars and Stins (*Roelofs, Hagenaars & Stins, 2010*).

## Experimental setting

The experiment took place in a dimmed experimental room. During the experiment participants stood on the stabilometric force platform for the majority of the time while the experimenter was present in the same room but out of sight.

## Procedure

The experimental session lasted approximately 120 minutes and consisted of a preparation and calibration phase, an acquisition phase, and a generalization phase. Figure 1 provides a

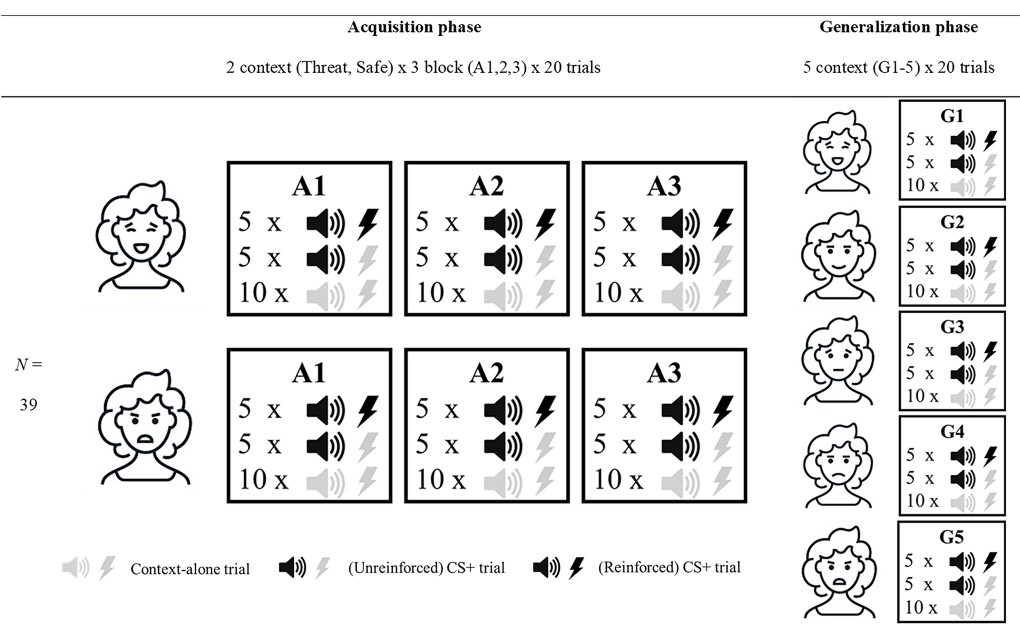

**Figure 1  Overview of the experimental design.** Overview of the three acquisition (A1-3) and five generalization blocks (G1-5). All blocks consisted of 20 trials, 10 of which had no auditory cue or pain-US (context-alone trials). The 10 remaining trials had an auditory cue (CS+ trials), of which 5 also involved the presentation of an electrocutaneous stimulus. Acquisition blocks were run in the threatening (represented by angry facial stimulus) and safe (represented by happy facial stimulus) social context. Generalization refers to the generalization contexts (G1-5) with varying degrees of social threat (17%, 33%, 50%, 67% and 83%) during the generalization blocks. Note that the facial stimuli used in this figure are merely symbolic, and that the order of the threatening and safe context in the acquisition phase, as well as the order of the generalization contexts, was intermixed and counterbalanced across participants.

more detailed overview of the acquisition and generalization phases, which we subsequently refer to as the experimental phases. A within-subject design was employed, meaning that all participants ran through both threatening and safe contexts. Note that the generalization phase was exploratory. Considering the high drop out in this phase the findings are unreliable and thus should be interpreted with caution. We decided to include a full description of the methods and results of the generalization phase in the supplementary material only rather than in the main manuscript to facilitate clarity (see Supplemental Information 3).

### Preparation and calibration phase

After participants arrived at the lab they were informed about the study orally and in writing. The participants were led to believe that the study concerned the effects of different kinds of distractors (auditory, visual and sensory) on balance (i.e., standing still on the platform). They were informed that painful electrocutaneous stimuli (pain-USs) would be administered during the experiment. After the participants provided informed consent, the equipment for the heart rate measurement was attached. Subsequently, the calibration of the electrocutaneous stimulus was performed. Afterwards the use of the platform was

explained, and participants removed their shoes and attached the safety harness. They were instructed to stand centrally on the platform with their arms hanging alongside their body, to move as little as possible during the experiment, and to watch the computer screen in front of them. Lastly, the electrodes for the pain-US administration were reattached and the safety harness was connected to the platform. Participants then read the following task instructions on the computer screen: "Please stand still on the platform and try to move as little as possible. You will be presented with several stimuli. Please focus on the center of the screen during the whole experiment. Sometimes you will be presented with sounds. These sounds might be followed by an electrocutaneous stimulus."

### Experimental phase

Facial stimuli were presented in six blocks (3 blocks with angry and 3 blocks with happy facial stimuli). Henceforth, these blocks will be referred to as A1 to A3 (also see Fig. 1). The order of the block presentation was intermixed, and the order of stimuli within each block was randomized. Each block consisted of 20 images of one type of emotional expression. During each trial the facial stimulus was presented for 3 s. In 50% of the trials (10 per block) an auditory cue was presented 1.5 s after the start of the trial (CS+ trials). In half of these trials (5 per block), the auditory cue would be followed by an electrocutaneous stimulus 1.3 s after the auditory cue (i.e., 50% reinforcement rate). The five trials with electrocutaneous stimuli would last for 5 s instead of 3 s to allow time for recovery. In the remaining 50% of trials (10 per block), no auditory cue or electrocutaneous stimulus was presented (context-alone trials). Thus, each block would take 70 s in total to complete.

After each block the safety harness was unhooked and the electrodes disconnected. The participant stepped off the platform and sat down at the table to fill in several questionnaires (see Outcome measures). This break would take approximately 5 minutes between blocks. Afterwards the same procedure was repeated for the remaining blocks.

## Outcome measures
### Self-report measures

*Pain-US expectancy.* In order to assess whether differential learning occurred and as a proxy for pain-related fear (*Boddez et al., 2013*), pain-US expectancy ratings were collected. Participants were presented with the following questions: (1) "*During the last block, how much did you expect that a face together with a tone would be followed by an electrical stimulus?*" and (2) "*During the last block, how much did you expect that a face without a tone would be followed by an electrical stimulus?*". Both questions were rated on an 11-point Likert scale (range 0-10) with labels "not at all" to "very much".

*Retrospective pain intensity and unpleasantness.* At the end of each block participants were asked the following questions: (1) "*How painful did you find the painful stimuli in the last block?*" and, (2) "*How unpleasant did you find the painful stimuli in the last block?*" which they rated on an 11-point Likert scale ranging from 0 to 10. The anchors were (1) "not painful at all" and "very painful" and (2) "not unpleasant at all" and (10) "very unpleasant".

*Retrospective affective valence, arousal and control of the facial stimuli and auditory cue.*
After each block, participants rated valence, arousal and sense of control of the facial stimuli using the Self-Assessment Manikin scale (SAM) (*Bradley & Lang, 1994*) consisting of 5 pictographs. Participants retrospectively rated how they felt when exposed to the facial stimuli. All responses were scored from 1 (very unhappy/not at all aroused/no control) to 5 (very happy/very aroused/full control). Similarly, at the beginning of the experiment (before the first block), and after the last block participants rated the auditory cue on these three measures.

*Perceived threat and pleasantness of facial stimuli.* After each block, participants answered the following questions: (1) "*How threatening did you find the facial stimuli in the last block?*" and (2) "*How pleasant did you find the facial stimuli in the last block?*" on an 11-point Likert scale ranging from 0 (not threatening at all/not pleasant at all) to 10 (very threatening/very pleasant). These measures were included as a manipulation check for the social context manipulation.

*State-trait anxiety inventory.* As in the study by *Roelofs, Hagenaars & Stins (2010)*, trait anxiety was measured by the trait version of the Spielberger's State-Trait Anxiety Inventory (STAI-T). This questionnaire assesses how anxious participants feel in general, using a scale ranging from 1 (not at all) to 4 (very much) (e.g., "*I feel nervous and restless.*"), and the scale has been shown to have high internal consistency, satisfactory test-retest reliability and concurrent validity with other anxiety measures (*Spielberger, Gorsuch & Lushene, 1970*). Mean anxiety scores in our sample were 37.61 (SD = 9.68, range = 24–66).

## Behavioral outcomes
### Body sway
Body sway was measured as an index of postural mobility (*Krampe, Smolders & Doumas, 2014*; *Roelofs, Hagenaars & Stins, 2010*). For each trial, the mean position of the center of pressure (COP) in the anterior-posterior (AP) and the medio-lateral (ML) directions was calculated. Referencing this mean, the standard deviation of the COP in the AP direction was computed and used as an index of variability in body sway. The standard deviations were then averaged across each block and stimulus type (CS/context-alone), excluding all trials where the pain-US was administered (25% of trials). In line with several earlier studies, we focused on variations in the AP-direction in our analyses, as these are more susceptible to affective modulations (*Hagenaars, Roelofs & Stins, 2014*; *Niermann et al., 2018*; *Niermann et al., 2015*; *Roelofs, Hagenaars & Stins, 2010*).

### Painful facial expression
A high definition webcam (Model c525, Logitech) was used to capture participants' facial expression. The webcam was installed right above the stimulus screen and adjusted per participant to capture the whole face during the experiment. Video tapes of each participant were rated using the Facial Action Coding System (FACS) (*Ekman & Friesen, 1978*), a fine-grained anatomically based system that is considered the criterion standard when decoding facial expressions, including the facial expression of pain (*Prkachin, 2009*).

Six facial action units which have been found to most reliably indicate pain are brow lower, eye squeeze, eye squint, nose wrinkle, check raiser and upper lip raise (*Kunz & Lautenbacher, 2014*; *Peeters & Vlaeyen, 2011*; *Prkachin, 2009*; *Prkachin, 1992*) were rated by the third author (TL), who was trained by the first author. The first author (KK), who is a certified FACS coder, also independently rated a randomly selected 20% subset of all video fragments. Each video fragment consisted of four-second segments capturing one second prior to, and three seconds after administration of the pain-US. Each second of the four-second interval was coded using a software program enabling the rater to view and review each second at normal rate and at a rate of one-tenth of a second. For each time interval, a mean score per second for each of the six facial actions was calculated. A total score was calculated by summing these mean scores per participant (*Caes et al., 2012*). Reliability was calculated according to the formula given by *Ekman & Friesen (1978)* which assesses the proportion of agreement on actions recorded by two coders relative to the total number of actions coded as occurring by each coder. Inter-rater reliability was satisfactory and ranged from .89 to .95 across all facial actions.

### Heart rate

The Polar belt and transmitter supported recording and processing of R-R intervals at a frequency of 1,000 Hz and 2.4 GHz transfer between the belt and heart rate monitor. The portable belt was attached around the chest at the height of the sternum. Unfortunately, the Polar did not make it possible to distinguish between individual trials so heart rate responses were averaged across blocks.

## Statistical analyses and data reduction

Paired-samples *t*-tests were carried out to compare the level of arousal, control and valence after hearing the auditory cue (CS) before and after it was paired with the painful stimulus. As a manipulation check for the social context manipulation, paired-samples t-tests were conducted to compare the valence, arousal, control, threat, and pleasantness ratings in response to the happy and angry facial stimuli. To this end, all ratings were averaged separately for the safe and threatening blocks.

Pain-US expectancy ratings and body sway were analyzed using repeated measures (RM) ANOVAs with 2 [Stimulus type (CS+/context-alone)] × 2 [Context (threat/safe)] × 3 [Block (A1, A2, A3)] as within-subject factors. Separate 2 [Context (threat/safe)] × 3 [Block (A1, A2, A3)] RM ANOVAs were run to examine the effects of social context on heart rate, facial pain expression and pain ratings (pain intensity and pain unpleasantness). In the analyses of heart rate, painful facial expression and body sway, centered STAI-T scores were included as a covariate.

To deal with frequent violations of the assumption of sphericity, multivariate analyses were run. Pillai's trace estimates and the effect size indication $\eta_p^2$ are reported. Planned comparisons were carried out to test our a priori hypotheses. Holm Bonferroni corrections were used to correct for multiple testing (*Holm, 1979*). All statistical analyses were run using IBM SPSS 20 (SPSS Inc, Chicago, IL). An alpha level of .05 was used for all statistical tests. We report how we determined our sample size, all data exclusions (if any), all manipulations, and all measures in the study (*Simmons, Nelson & Simonsohn, 2012*).

## RESULTS

### Dropout

The experimental procedure was quite demanding for the participants and 6 participants stopped their participation before the end of the experiment because they indicated that they started to feel unwell and as a precaution the experimenter decided to terminate the experiment. From the initial 39 participants, 33 participants (84.6%) completed the full experiment including the generalization phase, and 36 participants (92.3%) completed only the acquisition phase which is reported in this manuscript. The three (7.7%) remaining participants stopped their participation during the acquisition blocks. All subsequent analyses are run on the 36 participants who completed the acquisition phase unless otherwise stated.

### Manipulation checks

#### Retrospective affective valence, arousal and control of the auditory cue

There was no significant difference in reported level of arousal, Mbefore $= 2.50$, $SD = .97$; Mafter $= 2.42$, $SD = .10$; $t$ (35) $= .400$, $p = .69$, and only a slight trend with regard to valence, Mbefore $= 2.86$, $SD = .97$; Mafter $= 2.61$, $SD = .76$; $t$ (35) $= 1.86$, $p = .07$, after hearing the auditory cue (CS) before and after the experiment. However, the reported level of control after hearing the auditory cue (CS) before and after the experiment significantly decreased, Mbefore $= 4.14$, $SD = 1.40$; Mafter $= 3.61$, $SD = 1.40$; $t$ (35) $= 2.08$, $p = .05$.

#### Retrospective perceived threat, pleasantness, affective valence, arousal and control of the facial stimuli

As anticipated, angry faces were rated as less pleasant, Mangry $= 2.82$, SD $= 1.45$; Mhappy $= 6.02$, SD $= 1.88$, $t$ (35) $= -7.76$, $p < .01$, and more threatening, Mangry $= 5.81$, SD $= 1.69$; Mhappy $= 2.36$, SD $= 2.01$, $t$ (35) $= 9.05$, $p < .01$, than happy faces. Furthermore, participants indicated that they felt more unhappy (lower scores indicate more unhappiness), Mangry $= 2.28$, SD $= .57$; Mhappy $= 3.31$, SD $= .76$; $t$ (35) $= -7.74$, $p > .01$, while looking at the angry faces and also rated them as more arousing, Mangry $= 2.96$, SD $= .73$; Mhappy $= 2.57$, SD $= .72$; $t$ (35) $= 3.93$, $p < .01$, than happy faces. There was no significant difference in reported level of control, Mangry $= 2.53$, SD $= .84$; Mhappy $= 2.65$, SD $= .78$; $t$ (35) $= -1.10$, $p = .28$, between angry and happy faces. These results indicate that the social context manipulation was successful.

### Pain-US expectancy ratings

We found support for differential acquisition: Participants increasingly expected the pain-US following the presentation of the CS, but not when the context was presented alone, Stimulus type x Block, $F$ (2, 34) $= 8.79$, $p < .01$, $\eta_p^2 = .34$, (see Fig. 2). However, the magnitude of the differential acquisition effect did not differ between the two contexts, Stimulus type $\times$ Block $\times$ Context, $F$ (2, 34) $= 1.17$, $p = .32$, $\eta_p^2 = .07$. In contrast, pain-US expectancy ratings were overall higher in the threatening context, $M_{threat} = 6.70$, $SD_{threat} = 1.85$, compared to the safe context, $M_{safe} = 6.01$, $SD_{safe} = .2.21$, main effect of Context: $F$ (1, 35) $= 5.87$, $p = .02$, $\eta_p^2 = .14$.

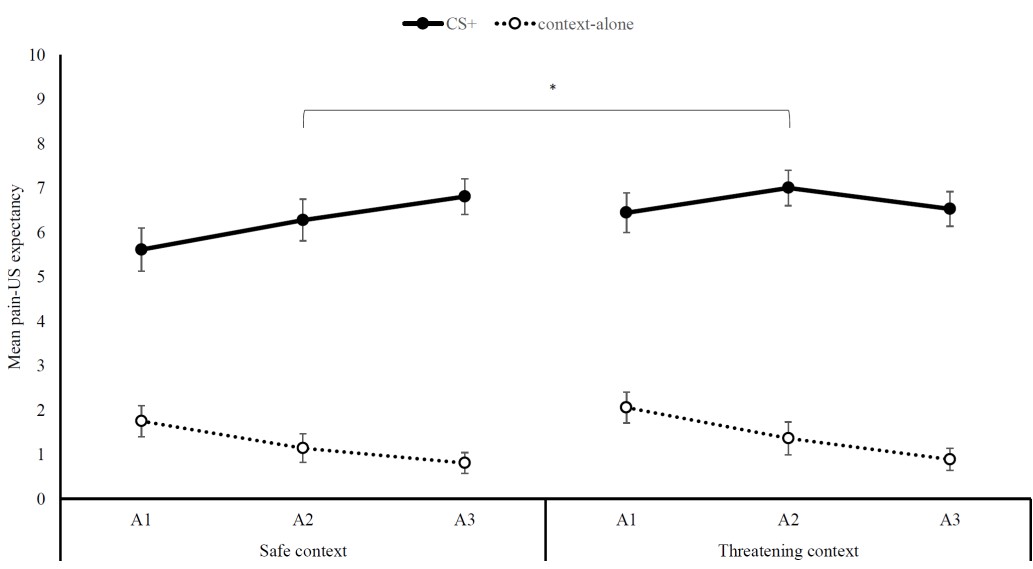

**Figure 2  Pain-US expectancy ratings.** Mean self-reported pain-US expectancy ratings (+ SEs) per block (A1-3) for trials with (CS+) and without (context-alone) the auditory cue, separately for the safe and threatening context. Note –SE = standard error term based on mixed analysis estimates; *, $p < .05$.

## Pain intensity and unpleasantness ratings

Pain intensity ratings did not differ across blocks, main effect of Block: $F (2, 34) < 1$, $p = .64$, $\eta_p^2 = .03$, or between contexts, main effect of Context: $F (1, 35) < 1$, $p = .68$, $\eta_p^2 < .01$, and there was also no interaction between block and context, Block × Context interaction: $F (2, 34) = 1.03$, $p = .37$, $\eta_p^2 = .06$. The same was true for pain unpleasantness ratings, Block: $F (2, 34) < 1$, $p = .42$, $\eta_p^2 = .05$; Context: $F (1, 35) < 1$, $p = .56$, $\eta_p^2 = .01$, Block × Context interaction: $F (2, 34) < 1$, $p = .96$, $\eta_p^2 < .01$.

## Hypothesis 1: does social threat modulate body sway in response to pain?

[1] The results of these analyses remained the same even when the outliers were included.

Two participants were identified as outliers because of their excessive movements, as reflected in $Z$-scores greater than 4 on the body sway measures.[1] Consequently, we decided to remove these participants from the analyses. Thus, the following analyses were run on 34 participants. The four-way interaction was not significant, Context × Block × Stimulus × Trait anxiety, $F (2, 30) < 1$, $p = .74$, $\eta_p^2 = .02$, and neither were any of the other interaction or main effects. There was a slight trend showing less body sway during CS trials compared to context-alone, Stimulus, $F (1, 31) = 3.03$, $p = .09$, $\eta_p^2 = .09$, but contrary to our hypothesis we found no evidence for any effect of social context or painful stimuli on body sway.

## Hypothesis 2: does social threat reduce heart rate?

Because of technical difficulties with the heart-rate monitor, recordings of three participants were missing and hence the following analyses are run on 33 participants. With regard to heart rate, a significant three-way interaction emerged, Block × Context × Trait anxiety, $F (2, 30) = 3.68$, $p = .04$, $\eta_p^2 = .20$ (see Fig. 3). We then evaluated context differences
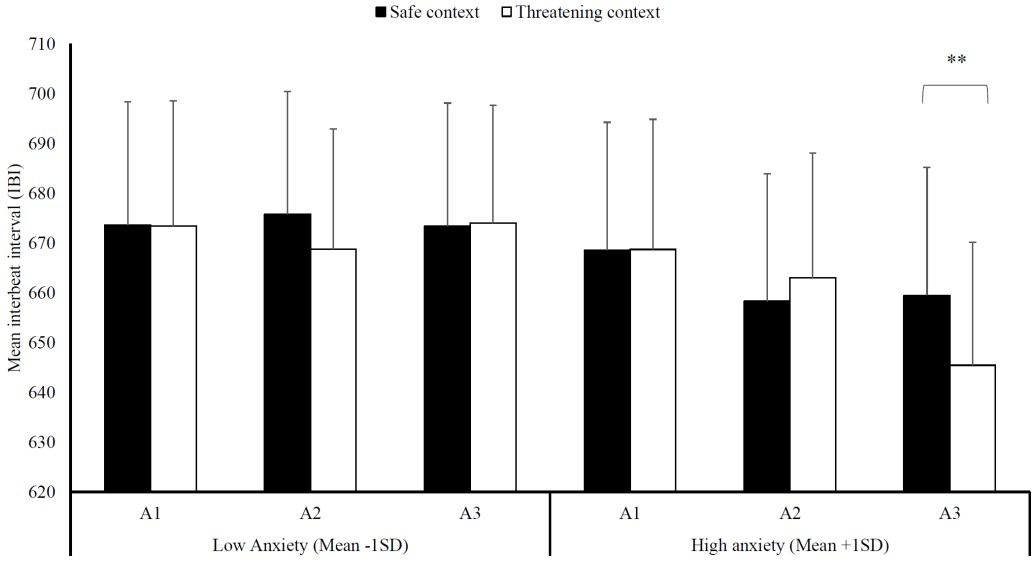

**Figure 3** **Heart-rate interbeat interval.** Mean interbeat interval (+ SE) per block (A1-3) in the safe and threatening context, separately for individuals low in trait anxiety (mean -1SD) and high in trait anxiety (mean +1SD). Note −SE = standard error term based on mixed analysis estimates; **, $p < .01$.

per block for high (mean STAI-T + 1SD) and low levels of trait anxiety (mean STAI-T −1SD) using planned comparisons. Note that this analysis models the interaction of block and context at different levels of the covariate (+1 SD and −1 SD) and thereby avoids the power loss associated with a median split. The model is thus based on the whole sample [see 27 for use of a similar analysis].

While there were no differences between a threatening and a safe social context in any of the blocks at low anxiety levels, $F(1, 31) < 1$, $p = .89$, $\eta_p^2 < .01$, there was a difference in heart rate at high anxiety levels. Specifically, the threatening social context, $M_{threat} = 645.47$, $SD_{threat} = 141.49$, led to reduced heart rate in the very last block compared to a safe social context, $M_{safe} = 659.49$, $SD_{safe} = 147.64$, $F(1, 31) = 9.92$, $p < .01$, $\eta_p^2 = .24$.

## Hypothesis 3: does social threat reduce facial pain expression?

Because of technical difficulties recordings of three participants were missing and hence the following analyses are run on 33 participants. Regarding facial pain expression, no significant three-way interaction emerged, Block × Context × Trait anxiety, $F(2, 30) = 1.26$, $p = .30$, $\eta_p^2 = .08$. As expected, we found a significant effect of social context that was moderated by trait anxiety, Context × Trait anxiety, $F(1, 31) = 5.35$, $p = .03$, $\eta_p^2 = .15$, (see Fig. 4). Subsequently, we evaluated simple effects of social context at high anxiety levels (1 SD above mean STAI-T score) and low anxiety levels (1 SD below mean STAI-T score). In line with our hypotheses, we found that there is less facial pain expressions in the threatening context at high anxiety levels, $M_{threat} = 2.62$, $SD_{threat} = 5.27$, compared to the safe context, $M_{safe} = 3.55$, $SD_{safe} = 5.56$, $F(1, 31) = 4.89$, $p = .04$, $\eta_p^2 = .14$. In contrast, there was no difference between the two contexts at low anxiety levels, $M_{threat} = 3.56$, $SD_{threat} = 4.91$; $M_{safe} = 3.09$, $SD_{safe} = 5.21$, $F(1, 31) = 1.24$, $p = .27$, $\eta_p^2 = .04$.

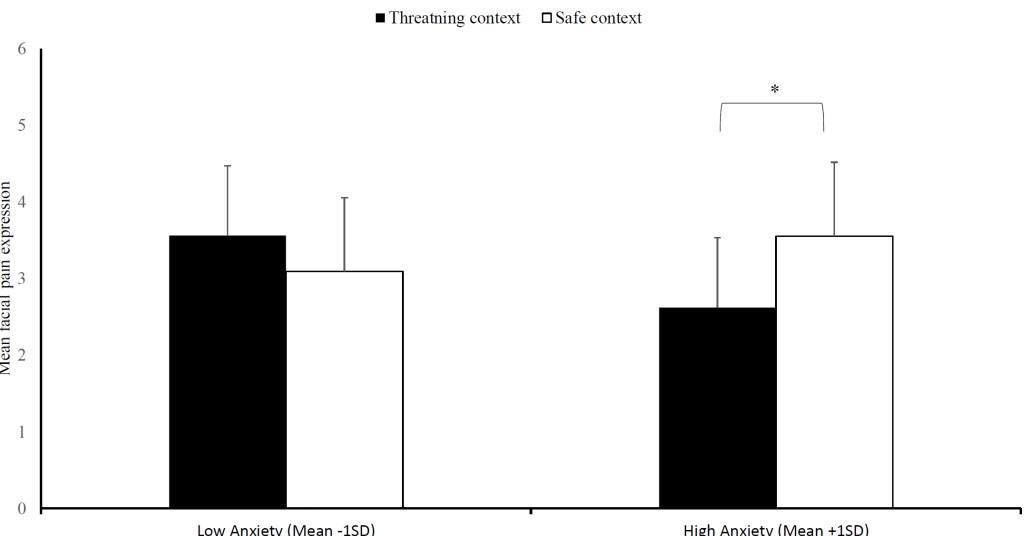

**Figure 4 Facial pain expression.** Mean facial pain expression (+SE) in the safe and threatening context, separately for individuals low in trait anxiety (mean -1SD) and high in trait anxiety (mean +1SD). Note – SE = standard error term based on mixed analysis estimates. *, $p < .05$.

## DISCUSSION

The current study aimed to explore the effects of a threatening versus a safe social context on freezing responses to pain, hypothesizing that (1) the anticipation of pain is associated with reduced body sway, and that this response is more pronounced in a threatening social context compared to a safe social context and especially so for highly anxious individuals, (2) that a threatening social context would induce bradycardia, and (3), reduce painful facial expression compared to a safe social context.

*First*, we found no support that the anticipation of pain leads to reductions in body sway (*Hypothesis 1*), or that body sway is modulated by social context. That is, we were not able to replicate the results reported by *Roelofs, Hagenaars & Stins (2010)*, who demonstrated that angry facial stimuli lead to reductions in body sway compared to happy facial stimuli. In addition, even though animal research indicates that the anticipation of painful electrocutaneous stimuli leads to freezing in rats (*De Castro Gomes & Landeira-Fernandez, 2008*), we did not see reductions in body sway in the anticipation of painful stimuli in humans. A possible explanation for this finding might relate to the proposed evolutionary function of freezing behavior. There is increasing evidence that freezing is an *active* preparation response rather than a *passive* orientation response (*Gladwin et al., 2016*; *Hagenaars, Oitzl & Roelofs, 2014*). That is, even though the present study and several earlier studies investigated freezing using passive viewing paradigms, one might expect freezing responses especially in situations where action is possible to respond to threat (e.g., a situation where one could avoid threat) rather than in a situation of helplessness as was the case here. A recent study by *Gladwin et al. (2016)* indeed found that freezing in the form of reduced body sway and bradycardia was strongly related to the possibility to respond. That
is, freezing was shown during a preparatory period in a virtual shooting simulation when participants had the possibility to respond to a possible attack in comparison to a situation where they did not have that possibility. Another possible explanation for this finding is that painful electrocutaneous stimuli themselves had a substantial effect on body sway in this study, overshadowing possible effects of social context or pain anticipation on body sway. Since we administered electrocutaneous stimuli in both contexts, we might have induced overarching contextual threat that led to freezing independent of social context. In other words, we did not create a truly *safe* context as a comparison. The addition of a baseline condition without painful stimuli might solve this problem. Another possible explanation might be that in contrast to the study by *Roelofs, Hagenaars & Stins (2010)*, all participants wore a safety harness during the study which might have affected mobility itself or acted as a safety signal which reduced overall threat perceptions during the study.

*Second*, in regard to *Hypothesis 2* predicting that a threatening social context would lead to reductions in heart rate (bradycardia) compared to a safe social context, we only found a highly specific effect: Social threat led to bradycardia at high anxiety levels and only so in the last block. So while these findings partly replicate the findings by *Roelofs, Hagenaars & Stins (2010)*, the effect of social threat seems to be rather specific for highly anxious individuals. We also observed an overall decline in heart rate across blocks at high anxiety levels (independent of social context) but not at low anxiety levels which is in line with earlier research in rodents (*Hagenaars, Oitzl & Roelofs, 2014*), and a similar time-course has been demonstrated in response to affective films (*Hagenaars, Roelofs & Stins, 2014*). This long-lasting time course demonstrates that the current paradigm might indeed cause freezing, a sustained defensive response, rather than just short-lasting orienting responses. In sum, as freezing in humans and animals is commonly defined as a reduction in mobility *and* bradycardia (*Hagenaars, Oitzl & Roelofs, 2014*; *Hagenaars, Roelofs & Stins, 2014*; *Roelofs, Hagenaars & Stins, 2010*), we can conclude that we found no evidence for freezing in the current study.

*Third*, mirroring the heart rate findings, we also found an effect of social threat on pain expression (*Hypothesis 3*), but again limited to high anxiety levels. A threatening social context led to reduced facial pain expression in comparison to the safe social context at high anxiety levels. These findings support an evolutionary account proposed by *Williams (2002)*, which proposes that it might be disadvantageous to express pain in a threatening social environment as it could indicate vulnerability which might be exploited by adversaries. Thus far, three empirical studies have supported this account (*Karos et al., 2019*; *Peeters & Vlaeyen, 2011*; *Williams et al., 2016*) and the current study also supports this idea, except that this effect was limited to highly anxious individuals.

It is still a question of debate whether these changes in facial pain expression might reflect conscious control or automatic processes (or both). While it has been argued that facial pain expression is more susceptible to conscious control than other pain behaviors (*Kunz, Rainville & Lautenbacher, 2011*; *Prkachin, 2009*; *Prkachin, 1992*), the present findings might lend some support for automaticity as well. While an earlier study found inhibition of facial pain expression in response to real-life others (*Peeters & Vlaeyen, 2011*), we demonstrated this inhibition for the first time in response to inanimate facial stimuli. One possible

explanation is that these stimuli indeed acted as social threat cues and activated hard-wired, automatic reactions in facial pain display as proposed by evolutionary accounts of pain expression (*Williams, 2002*). It is also interesting to note, that this inhibition was only observed in highly anxious individuals. That is, highly anxious individuals were more sensitive to a threatening social context than low anxious individuals and consequently, demonstrated defensive responding in response to passive facial stimuli. This finding is rather surprising, considering that anxiety has been associated previously with a lack of inhibitory capacity (*Ansari & Derakshan, 2011*). This finding may be clinically relevant, as patients with pain complaints might inhibit facial expression of pain if they do not feel safe (e.g., do not trust a healthcare professional). One possible consequence of this might be that the pain of the patient is underestimated by others, as is commonly the case in clinical practice (*Karos et al., 2019*; *Seers et al., 2018*).

Moreover, reductions in painful facial expression were present independently from freezing response (i.e., reductions in body sway and bradycardia), suggesting that these are likely independent constructs. While reductions in facial pain expression could be explained by general body immobility as demonstrated in the context of freezing, the current study demonstrates that inhibition of facial pain expression can occur independently of observed freezing behavior.

*Fourth*, we also explored the effects of a threatening social context on pain expectancy, pain intensity and pain unpleasantness. We found no support that social threat directly affects pain intensity or unpleasantness ratings. There is limited support that a threatening social context in the form of intentional pain by others might directly affect pain intensity ratings (*Gray & Wegner, 2008*; *Karos et al., 2019*; *Krahé et al., 2013*) but studies using facial stimuli have thus far not shown similar results (*Karos, Meulders & Vlaeyen, 2015*). Instead, an earlier study using angry and happy facial stimuli to manipulate social context has found that fear learning was facilitated in a threatening social context (*Karos, Meulders & Vlaeyen, 2015*). In the present study, we found that overall pain expectancy ratings were higher in the threatening social context compared to the safe social context. That is, participants expected more pain in the threat context independently of the presence (or absence) of the CS. This finding seems to demonstrate contextual fear (i.e., hypervigilance), which is usually observed in a context where cued learning is impossible (e.g., an unpredictable US) (*Meulders, Vansteenwegen & Vlaeyen, 2011*), or when individuals fail to learn the CS-US association (*Baas et al., 2008*). In the current study, participants expected more pain in the threatening social context *despite* learning the CS-US association. In other words, a threatening social context led participants to predict that the occurrence of threatening stimuli (i.e., pain) would be more likely.

There are a few limitations to this study that should be noted. First, for practical reasons we were only able to measure heart rate across blocks rather than per trial, which would have been desirable to distinguish between pain and non-pain trials. This would allow a direct comparison with the body sway data, which could be evaluated on a trial-by-trial basis. Second, the current study suffered from drop out which means that some of the analyses (specifically, the heart-rate and pain expression analyses) are not as robustly powered as we would have hoped and that the results presented here should be interpreted

with caution. It should be noted that the participants who completed the full experiment and those who dropped out before the end did not differ in regard to anxiety, pain catastrophizing, fear of pain, or gender. Third, there are several variables that might affect postural stability which were not recorded in the present study (e.g., physical activity levels, sleep, use of drugs, alcohol, or nicotine). Some of these variables might have overshadowed the manipulation used in this study and should therefore be assessed and controlled for in future studies. Fourth, we found large variability in heart-rate and facial pain expression scores which suggests for large inter-individual variability. Even though this is in line with earlier studies (*Karos et al., 2019*; *Shaffer & Ginsberg, 2017*), future studies might attempt to control for confounding variables (e.g., physical activity, gender, age) to reduce this variability and increase statistical power. Fifth, the current study recruited pain-free controls and made use of an experimental pain induction method. Consequently, any extrapolations of the findings to patients with chronic pain are premature at this stage. However, the basic assumption with this fundamental research is that the underlying mechanisms in healthy populations under certain manipulations can inform us about what is happening in patient populations. Lastly, the current study did not have a predefined and publicly available analysis plan prior to the start of the study. Even though this was not common practice at the time that this study was conducted, future studies should preregister their plan of analysis and hypotheses to further limit potential bias and facilitate transparency in science (*Lee et al., 2018*; *Nosek et al., 2015*).

## CONCLUSIONS AND FUTURE DIRECTIONS

In conclusion, the current study provides preliminary evidence that social context modulates pain-relevant processes, especially in highly anxious individuals. Social threat increased overall expectation of pain and reduced facial pain expression. In addition, increased bradycardia in the threatening social context was restricted to a single block and only present in highly anxious participants. In contrast, we did not find freezing in response to painful stimuli and also no effect of social context on body sway. This research relates to earlier studies showing impaired safety learning and excessive avoidance behavior in highly anxious individuals, and might add to our understanding of how chronic pain complaints develop and are maintained (*Meulders, Meulders & Vlaeyen, 2014*; *Meulders & Vlaeyen, 2013*). In addition, the present research further highlights the importance of social context in the study and understanding of pain (*Karos et al., 2018a*).

There are several possible avenues for future research. First, more research is needed on the effects of acute and chronic pain on freezing parameters. To this end, the effect of experimental pain induction methods and anticipation of pain on freezing responses in humans should be investigated in isolation. In addition, freezing responses in patients with chronic pain should be compared to pain-free controls to investigate possible differences in defensive responding. There is some evidence for maladaptive freezing responses in other psychopathologies such as PTSD (*Niermann, Figner & Roelofs, 2017*) and in rats with chronic pain (*Lamana et al., 2018*), but research in humans is absent. It is well established that patients move differently when they are in pain (*Hodges & Tucker, 2011*) and even

the anticipation of pain changes movement (*Karos et al., 2017*). If chronic pain patients do show signs of maladaptive freezing responses, as they do with avoidance behavior, longitudinal research is needed to evaluate the role of this process in the etiology of chronic pain complaints. Second, more research into the modulation of defensive behaviors (e.g., freezing, pain-related avoidance behavior) by social context is warranted. One possibility is the use of more dynamic and ecologically valid social manipulations such as virtual reality, video materials or actual social interactions using confederates (*Karos et al., 2019*; *Karos et al., 2018*).

## ACKNOWLEDGEMENTS

The authors thank Jeroen Clarysse, Mathijs Franssen and Professor Ralf Krampe for their technical support.

### Funding

The contribution of Johan Vlaeyen was supported by the "Asthenes" long-term structural funding Methusalem grant by the Flemish Government, Belgium. The contribution of Ann Meulders was supported by a Senior Research Fellowship of the Research Foundation Flanders (FWO-Vlaanderen), Belgium, (grant ID: 12E3717N) and a Vidi grant from the Netherlands Organization for Scientific Research (NWO), The Netherlands (grant ID 452-17-002). Kai Karos is a post-doctoral researcher supported by the Research Foundation Flanders (FWO-Vlaanderen), Belgium (grant ID = 1111015N and 1244820N). The funders had no role in study design, data collection and analysis, decision to publish, or preparation of the manuscript.

### Grant Disclosures

The following grant information was disclosed by the authors:
Flemish Government, Belgium.
Senior Research Fellowship of the Research Foundation Flanders (FWO-Vlaanderen), Belgium: 12E3717N.
Netherlands Organization for Scientific Research (NWO), The Netherlands: 452-17-002.
Research Foundation Flanders (FWO-Vlaanderen), Belgium: 1111015N, 1244820N.

### Competing Interests

The authors declare there are no competing interests.

### Author Contributions

- Kai Karos conceived and designed the experiments, performed the experiments, analyzed the data, prepared figures and/or tables, authored or reviewed drafts of the paper, and approved the final draft.
- Ann Meulders and Johan W. Vlaeyen conceived and designed the experiments, authored or reviewed drafts of the paper, and approved the final draft.

- Tine Leyssen performed the experiments, authored or reviewed drafts of the paper, analysis of pain expression data, data collection, and approved the final draft.

## Human Ethics

The following information was supplied relating to ethical approvals (i.e., approving body and any reference numbers):

The experimental protocol was approved by the Social and Societal Ethics Committee of the KU Leuven (Belgium) (registration number = S-56678).

## Data Availability

The data that all statistical analyses were run on (including all self-reported data), raw heart rate data, raw body sway data, and raw pain expression ratings and reliability scores are available as Supplementary Files.

All facial stimuli used in this study come from the Karolinska Directed Emotional Faces (KDEF) which is available at https://www.kdef.se/.

## Supplemental Information

Supplemental information for this article can be found online at http://dx.doi.org/10.7717/peerj.10094#supplemental-information.

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
