# Peer review of "Freeze-like responses to pain in humans and its modulation by social context"

_PeerJ, doi:10.7717/peerj.10094_

## Round 0.1 · original submission · Major Revisions

You will find attached very comprehensive comments from the reviewers. One reviewer has more concerns with the methods and the ability to draw clear conclusions on the basis of your results. I would encourage you to pay particular attention to responding those points. There are also easily modifiable typographical and grammatical changes required. Above, your email indicates that there is a good chance your article will be accepted. I would like to iterate that this is not guaranteed and that some of the reviewers' observations require considerable thought and may not be correctable. However, if you can address the concerns then I look forward to seeing your revised version.

Reviewer 1 ·

Basic reporting

The manuscript is well written and mostly easy to follow. There are some minor comments in the general comments section to assist with the readability. The main focus is on the reporting of the participant numbers and ensuring this is accurately described considering the high drop out number.

Experimental design

The experiment is well considered, the question is well articulated and matches with the methods, reporting and discussion.

Validity of the findings

Negative results are well accepted. There are some minor comments below regarding the interpretation and not overstating the findings.

Additional comments

Methods
1. Please explain why a medium effect size was used in the power calculation.
2. Line 139 make clear that the electrocutaneous stimulus was designed to be painful
3. Line 149 rephrase “receive stimuli of maximally this amplitude” to explain more clearly the effect of the amplitude on what was experienced by the participants.
4. Line 165 refers to table 1 and I was expected to see the preparation and calibration phrase included in the table. Please rephrase to explain more clearly that table 1 only refers to the acquisition and generalisation phase. Perhaps make clear the distinction between the preparation and calibration and that the experimental phase contains the acquisition and generalisation phase and this is how it is explained further down in the manuscript.
5. Line 168 the phrase absence of any findings indicates that no data was gathered, rephrase to explain more clearly that there was no data was not meaningful due to the high drop out rate. Very good that it is still reported in the supplementary material.
6. Line 183 refers to the task instructions please include these as a supplementary material or indicate that the instructions are available from the authors for replication.
7. Line 186 experimental phase: could the authors consider a more clear table or figure to visually guide the reader through the experiment. Table 1 attempts to do this however I feel it could be more clearly explained.
8. Line 189 it was not clear how it was decided which trials would have the auditory cue and of those trials with the auditory cue which would have the electrocutaneous stimulus.
9. Line 264 the reference to table 1 is confusing here, as to how table 1 would help the reader understand the heart rate recordings.
10. Statistical analysis: : Line 272 when describing the ANOVAs please explain the block references i.e. A1, A2, A3 and what these refer to. The references to A1, A2 and A3 are not included previously in the manuscript and it is not clear to the reader.
Results
11. Line 287 with regards to the participants who did not finish the experiment, the methods section states that 39 volunteered and that 7 participants stopped their participation and that 36 completed the full experiment. Please clarify and alter the text to explain how 36 participants still complete the full experiment when 7 dropped out from a starting number of 39.
12. In addition to clarify the point above, given the high dropout rate 7/39 (18% dropout) please describe the characteristics of the participants who dropped out and if they differed to the participants who stayed in on any of the outcome measures that were assessed. i.e. anxiety, gender.
13. Please also clarify the number of participants who were included in the final analysis. The numbers in the results indicate that 39 participants were included.
14. Line 294 with regards to reporting of the t tests and interpretation of the p values, please define in the methods section how you planned to interpret the results. You describe a “slight trend” regarding valence, and that for control you describe a significant decrease. The reader would assume that you set the significance level at 0.5 but please make this explicit. In addition, to reporting the p values please also include the mean difference of the pre and post scores where possible, unless the scientific editor please this is not necessary.
15. Line 339 it is stated that a planned comparison was used, however there is no further explanation for this. As a reader I was unclear of the reference that was used and why this was included. In the referenced article (27) were similar comparisons done? Please explain this.
16. For Hypothesis 2 in the results section please include how many participants were included in this analysis for the low and high anxiety groups.
17. Furthermore for hypothesis 2 with the reporting I feel it would be important to include the mean between group difference and if the large SD are not explained further on they I would ask that the large SD are explained somewhere in the text. It is hard to appreciate that difference in the between groups scores when the SD are so large.
18. Regarding figure 2 and 3, I appreciate that this is a common graph for this style of research however I would encourage the authors in the future to consider presenting the results in graphical form in such a way that allows the lower SD to also be presented and not just the upper level. Given the results in this paper I do not feel it is necessary to change the graphs but would encourage this be considered for future manuscripts.
19. Hypothesis three- my comments above regarding reporting the between group means and SD is also appropriate for hypothesis 3.
Discussion
20. Line 375 between “relate” and “the” it seems a word is missing
21. Nice interpretation of the results, link to previous research and possible explanations for the results.
22. Line 395 I would like the authors to consider rephrasing the term partial support. The description in the next line that the effect was specific would be a more accurate way to begin this paragraph and ensure the reader is not mislead.
23. I applaud the authors for their interpretation of the results in relation to hypothesis 2 in particular line 403 to 405
24. Would the authors please provide a possible explanation for the overall decline in heart rate across blocks for highly anxious individuals.
25. Line 417 replace colon with full stop.
26. Line 426- the extrapolation to clinical relevance is a large jump. I would prefer the authors to indicate that the findings “may” be clinically relevant not that the finding is clinically relevant.
27. Line 454 states that all participants were female however in the participants section it states that of the 39 participants 31 were female. Please correct this.
28. I would be interested for the authors to present their views on where this line of research could go in the future and what questions future studies should evaluate based on the findings of this research.
Conclusion:
29. Line 463 I would ask the authors to tone down the expression that bradycardia was especially pronounced and while you then state this was only in the anxious participants I still feel this overstates the findings. Please reword this statement.
30 I would encourage the authors in the future to predefine their analysis plan using an open access repository to make it clear what was decided prior to the analysis being conducted.

Reviewer 2 ·

Basic reporting

1. Your introduction needs more detail to provide a better justification for your project. I suggest that you improve the following elements:

• Which behavior changes are correlated with the transition from acute to chronic pain? From a psychological view, what are the differences between acute and chronic pain?
•More explanation on how a maladaptive responses can be linked to the social context should be added this section.
•Providing more quantitative comparison of the chronic pain prevalence and severity with adults living in different social environment would be relevant.
•The introduction would be strengthened by adding brief mention of other freezing indicators (Respiration, increased muscle tonus, reduced vocalization) and their related physiological explanations.

2. Figures may not reproduce well. I recommend that the authors darken the colors and thicken X and Y axes

Experimental design

1. A clear link between the proposed objectives, expected outcomes, and justification for the study have not been well established. It appears that the main thrust for the study is that no one has investigated these variables in patient with pain. It is not clear how these expected results, by using an electrocutaneous-induced acute pain model on healthy individual, will help inform future pain management protocols nor how the social context plays roles in chronic pain development and maintenance. Please explain why you used this study population and not chronic pain population.

2. Aims (Hypothesis) 1 and 2 propose to examine the impacts of threatening social context on specific outcomes in relation to anxious level. However, anxiety level was not an eligibility criteria. Did you have de same number of low anxious participants than the high anxious participants?

3. Since physical activity levels, sleep hygiene, energetic drinks have effects on postural stability, I would recommend to record this data of each participant.

4.The sample size calculation show adequate power, but I was unable to find the standard deviation of which parameter was using in this calculation.

5. Your stabilimetric force platform is not the most appropriate tool for obtaining COP derivation parameters (ellipse area of COP and the mean COP velocity in the anteroposterior (A/P) and mediolateral (M/L) directions). Only these COP parameters are sensitive and reliable postural control measures as published by Oliveira et al. 2019 (M.R. Oliveira et al. / Journal of Bodywork & Movement Therapies 23 (2019) 594e597). Please explain why you used this method. What are the psychometric properties of your measurement tool?

6. Please provide a complete electrocutaneous stimulation parameters as published by Geng et al (Geng et al. Journal of NeuroEngineering and Rehabilitation 2011).

7. Please add an explanation and references on your auditory cue parameters.

8. The social context method needs more details. Please explain how bias and variance were calculated. Why did you choose this method? Was it the most appropriate method for this issue (Virtual environments, dynamic or static face stimuli, photo-realistic faces, videos)?

Validity of the findings

1. How many participants were included in the groups «low anxious participant» and «high anxious participant»? Please explain de validity of the statistical analysis.

2. What were the sample characteristics (social environment, age, education level, physical activity level, initial COP values, anxiety, BMI)?

3. You need at least 36 patients to validate your statistical analysis. 39 participants were recruited, but 7 patients have dropped out before the end. 39 -7 = 32 and dropout rate of 18%. You wrote «A total of 36 (93.21%) completed the full experiment», is it a mistake? If you have 32 participants, please explain de validity of the statistical analysis. How dropout patients were handled in your statistical analysis?

4. Hodges et al. (P.W. Hodges, K. Tucker / PAIN 152 (2011) S90–S98 ) have shown how patients moving differently in pain. Explain how your sample population or pain model may induce a limitation in your results.

Additional comments

This study tests the proposal that pain stimuli can induce a freezing response in different social contexts. It finds correlations between pain anticipation, social context, body sway, anxiety and face expressions. This is an interesting study. However, in my opinion, this manuscript has a number of general and specific problems (major), ranging from basic grammar and readability, experimental design, statistical analysis, and validity of the findings. In several instances I also suggested to cite more relevant and recent literature. I do believe there is valuable data to be published from this research, but as it stands the manuscript is weak and unrefined. This is why I suggest a withdraw for the manuscript, as I believe the changes that need to be made to this manuscript are many and would be difficult for the authors to achieve in the time frame allowed. I hope that the following critiques will aid the authors in the refining of the manuscript to allow for re-submission in the future.

---

## Round 0.2 · Minor Revisions

I agree with the reviewer in that this paper seems much easier to follow now and the changes have improved it. I also agree that there remain two issues that should be addressed - a simply matter of wording and an acknowledgement of the contemporary standard of lodging protocols and plans prior to stages of hte study. The reviewer recommends two citations that I agree would justify the point. I hope you see these changes as useful and I look forward to seeing the next and hopefully final version!

Reviewer 1 ·

Basic reporting

LINE 236: The sentence with maximally in it does still not make sense: I think it would be better written: "After selecting the pain stimulus the participant was informed that (s)he would repeatedly receive stimuli of the maximum calibrated intensity during the remainder of the experiment."

Experimental design

I am now happy with the reporting of the results. Initially I had hoped for the mean differences, however on reflection on the nature of this research I think the information that the authors have provided is appropriate. It makes the findings much more transparent.

Validity of the findings

I am happy with the way the authors have interpreted the findings now.

Additional comments

Well done on the changes to the paper. I appreciate they were significant and time consuming changes. I hope you feel they improved the paper. From the perspective of the reader the paper is now much easier to follow and it is clear what happened with the participants along the way.

Regarding the final comment in my previous review about predefining the analysis plan, I appreciate that the authors understand the importance of this and acknowledge that this was not common practice when the study was designed. I feel that to help encourage others researchers that the authors should still make a comment in the discussion that not having a predefined and publicly available analysis plan is in fact a limitation and that future studies. A reference to a paper like the ones below would be appropriate:

Lee H, Lamb SE, Bagg MK, Toomey E, Cashin AG, Moseley GL. Reproducible and replicable pain research: a critical review. PAIN. Published online April 2018:1. doi:10/gf9dbc

Nosek BA, Alter G, Banks GC, et al. Promoting an open research culture. Science. 2015;348(6242):1422-1425. doi:10/gcpzwn

---

## Round 0.3 · accepted · Accept

Congratulations on this one - the reviewers were thorough and I think the paper is really nice. I am happy for it to proceed - an important addition to the field!